# Antimicrobial Proteins: Structure, Molecular Action, and Therapeutic Potential

**DOI:** 10.3390/pharmaceutics15010072

**Published:** 2022-12-26

**Authors:** Mohamed Hassan, Thomas W. Flanagan, Naji Kharouf, Christelle Bertsch, Davide Mancino, Youssef Haikel

**Affiliations:** 1Department of Endodontics, Faculty of Dental Medicine, Strasbourg University, 67000 Strasbourg, France; 2Department of Biomaterials and Bioengineering, INSERM UMR_S 1121, Biomaterials and Bioengineering, 67000 Strasbourg, France; 3Research Laboratory of Surgery-Oncology, Department of Surgery, Tulane University School of Medicine, New Orleans, LA 70112, USA; 4Department of Pharmacology and Experimental Therapeutics, LSU Health Sciences Center, New Orleans, LA 70112, USA

**Keywords:** antimicrobial peptide (AMP), multi-drug resistance (MDR), extracellular polymeric substances (EPSs), lipopolysaccharides (LPS), lipoteichoic acid (LTA)

## Abstract

Second- and third-line treatments of patients with antibiotic-resistant infections can have serious side effects, such as organ failure with prolonged care and recovery. As clinical practices such as cancer therapies, chronic disease treatment, and organ transplantation rely on the ability of available antibiotics to fight infection, the increased resistance of microbial pathogens presents a multifaceted, serious public health concern worldwide. The pipeline of traditional antibiotics is exhausted and unable to overcome the continuously developing multi-drug resistance. To that end, the widely observed limitation of clinically utilized antibiotics has prompted researchers to find a clinically relevant alternate antimicrobial strategy. In recent decades, the discovery of antimicrobial peptides (AMPs) as an excellent candidate to overcome antibiotic resistance has received further attention, particularly from scientists, health professionals, and the pharmaceutical industry. Effective AMPs are characterized by a broad spectrum of antimicrobial activities, high pathogen specificity, and low toxicity. In addition to their antimicrobial activity, AMPs have been found to be involved in a variety of biological functions, including immune regulation, angiogenesis, wound healing, and antitumor activity. This review provides a current overview of the structure, molecular action, and therapeutic potential of AMPs.

## 1. Introduction

The antimicrobial resistance (AMR) to available therapeutics is a serious healthcare problem that is mostly associated with the death of a specific portion of people worldwide. Moreover, current predictions indicate a significant increase in annual global death in the future [1].

AMR is a multifaceted health problem that represents a serious global threat. In addition, the traditional antibiotic pipeline is exhausted and unable to overcome the continuously developing multi-drug resistance. As consequence, the widely observed limitation of clinically utilized antibiotics has prompted researchers to find clinically relevant antimicrobial approaches. Accordingly, the discovery and development of alternative therapeutic approaches to overcome the widely reported AMR are urgently need. Antimicrobial peptides (AMPs) are group of small peptides, which are reported to play a crucial role in the host innate immunity against a broad spectrum of microorganisms, including bacteria (Gram-positive and Gram-negative), viruses, fungi, and parasites [1,2]. Accordingly, AMPs belong to the first-line defense of a host against invading pathogens. These AMPs are, in great part, amphipathic peptides with α-helical structures and β-sheets linked by disulfide bridges, extended loops, or cyclic configurations [2,3]. Thus, based on their broad-spectrum antimicrobial activity, killing potential, high selectivity, and low toxicity, AMPs have gained further interest from researchers and physicians as an alternative approach to the widely utilized antimicrobial agents, especially over the past two decades. The currently identified AMPs can be classified into different groups according to the information provided on the Data Repository of Antimicrobial Peptides (DRAMP) [http://dramp.cpu-bioinfor.org/ accessed on 1 August 2022]. AMPs are derived from six kingdoms (bacteria, archaea, protozoa, fungal, plants, and animals). As widely reported, AMPs are involved in a variety of biological functions, including immune regulation, angiogenesis, wound-healing, anti-inflammatory activities, and antitumor activity [3,4,5,6]. Furthermore, AMPs function as critical effectors in both innate and adaptive immunity. Thus, beyond their functional role as a link between innate and adaptive immunity, AMPs contribute to the resolution of inflammation [3,4]. The activation of the innate immune system by AMPs is recognized to be one of the key mechanisms that regulate AMP-mediated early clearance of infections. The immunomodulatory functions of AMPs are known to be very complex and involve various receptors, signaling pathways, and transcription factors [3,4]. AMPs can either directly or indirectly promote the recruitment of different immune cells, such as immature dendritic cells (iDCs), T lymphocytes, monocytes, eosinophils, and neutrophils, to the site of infection.

While the primary treatment of bacterial pathogens relies on established antibiotics, the development of multi-drug resistance (MDR) is ever evolving and changing [7,8]. In contrast to established antibiotics, AMPs have been approved for their therapeutic potential to overcome the multi-resistance of most microbial pathogens [5]. However, the clinical advantage of AMPs over currently available antibiotics resides in their mode of molecular action. Specifically, AMPs have been shown to kill microbial pathogens via a mechanism mediated by the destruction of plasma membranes and interference with intracellular components [2,7]. Despite the success of AMPs in clinical application, natural AMPs have doubtful properties that hinder their functional application [8]. However, the functional analysis of the chemical structure of AMPs may help to improve the molecular action and, subsequently, the therapeutic potential of AMPs. This review will discuss the structure, molecular action, and therapeutic potential of AMPs.

## 2. Sources and Structure of Antimicrobial Peptides

In addition to their distribution in all organisms, many if not all AMPs are evolutionarily conserved and derived from viral, bacterial, fungal, plant, and animal sources.

AMPs with viral sources include endolysins (lysins), virion-associated peptidoglycan hydrolases (VAPGHs), depolymerases, and holins, which are derived from bacteriophages [9].

AMPs with bacterial sources have been reported in several studies. For example, Bacillus strains have been shown to produce AMPs with promising inhibitory activity against *Shigella*, *Salmonella*, *E. coli*, and *Staphylococcus aureus* [10,11,12,13]. Furthermore, AMPs derived from Bacillus sp have been reported for their antimicrobial activity against *Staphylococcus aureus*, Alteromonas sp. strain CCSH174 and *Klebsiella pneumoni* [10,11]. Another example of bacterial AMPs includes those derived from *Propionibacterium jensenii* [14] and those isolated from Pseudomonas [12], which have been reported for their activity against *Shigella*, *Salmonella*, *E. coli*, and *Staphylococcus aureus* [15].

These AMPs with bacterial sources have been reported to be both ribosomally and non-ribosomally synthesized peptides.

Ribosomally synthesized bacterial AMPs are known as bacteriocins. Bacteriocins have been suggested as promising alternative approaches to the conventional small-molecule antibiotics. These bacteriocins can be divided into four classes. One of these classes is class I, which includes a group of AMPs that consist mainly of small peptides of 19–38 amino acids. The second class of the bacteriocins includes heat-stable AMPs, which are commonly synthesized as a prebacteriocin. The third class contains a group of large and heat-stable peptides, and the fourth class IV contains uniquely structured bacteriocins containing amino acids, lipids, or carbohydrates, in addition to being susceptible to lipolytic and glycolytic enzymes [9,16]. These AMPs are only active against bacteria, which are closely related to the producing strains but not their own producers [16].

Non-ribosomally synthesized AMPs of Gram-positive bacteria include cyclic lipopeptides, which are known as polymyxins, and linear peptides, which are known as tridecaptins. In contrast to Gram-positive-bacteria-derived AMPs, the majority of AMPs isolated from Gram-negative bacteria are common in *E. coli*, as well as in other species including Klebsiella spp. and *Pseudomonas* spp. [17]. These AMPs have limited activity against Gram-negative bacteria and can be classified into four classes, namely, colicins, colicin-like bacteriocins, microcins, and phage tail-like bacteriocins [17]. The classes of colicines are predominantly produced by *E. coli*. Although the class of colicin-like-bacteriocins is structurally and functionally similar to the colicins of *E. coli*, a number of other species, including *P. aeruginosa* and the Klebsiella genus, have been reported to produce colicin-like bacteriocins [18]. Furthermore, other bacteriocins such as microcins can be produced by Enterobacteriaceae, and they are active against phylogenetically close species [18]. The fourth class of bacteriocins includes Gram-negative-bacteria-derived AMPs, including phage tail-like bacteriocins [19]. This type of bacteriocin is characterized by its high molecular weight and cylindrical peptides [19].

Fungal AMPs are common AMPs, which are generally grouped into two main classes, fungal defensins and peptaibols [20].

Defensins are short, cysteine-rich peptides with different sources, including microorganisms, plants, and animals. Therefore, fungal-derived defensins are known as defensin-like peptides based on their high sequence and structural similarities. Although fungal AMPs are similar in their structure and peptide sequences, their activities against Gram-positive and/or Gram-negative bacteria and/or fungi differ [21].

Although there are different sources of AMPs, their numbers of amino acid residues range between 10 and 60 amino acids, and most of them are cationic with an average net charge of 3.32. In addition to cationic AMPs, there are also many anionic AMPs that contain several acidic amino acids, such as aspartic acid and glutamic acid.

Plant-derived AMPs are cysteine-rich peptides with broad-spectrum antimicrobial activity against bacteria, fungi, and viruses, and they possess immunomodulatory activities [22]. These AMPs are classified into various families based on their cysteine motifs, the arrangement of disulfide bridges, and sequence similarity. The most common members of plant-derived AMPs include α-hairpinin, defensins, hevein-like peptides, cyclic and linear knottin-type peptides, lipid transfer proteins, thionins, and snakins, in addition to unclassified cysteine-rich AMPs [22].

The AMPs with animal sources include invertebrate AMPs, fish and amphibian AMPs, reptile- and avian-derived peptides, and mammal-derived AMPs.

Invertebrate AMPs include those of insects, such as defensin and cecropin, mollusc AMPs (e.g., defensins), nematode AMPs (defensins), and horseshoe crabs (e.g., big defensins), in addition to invertebrate β-defensin and crustacean AMPs (e.g., crustins) [23]. Invertebrate AMPs are an integral component of humoral defense since the invertebrates lack an adoptive immune response when compared with those of the animal kingdom [23].

Vertebrates AMPs, particularly those of fish and amphibian origin, have been shown to play an essential role in defense responses to microorganisms. Although fish are a considerable source of several AMPs, such as cathelicidins, β-defensins, hepicidins, piscidins, and histone-derived peptides [24,25], amphibians are the largest source of AMPs among invertebrates. However, the most common amphibian AMPs include bombinins, buforin, cathelicidin, dermaseptins, esculentins, fallaxin, magainins, maximins, phylloseptins, phylloxin, plasticins, plasturins, pseudins, and ranateurins [25].

AMPs with reptile and avian sources belong to the members of the cathelicidin and defensin families [26]. Cathelicidins are small-sized AMPs secreted from macrophages and neutrophils upon their activation in response to infection. β-defensin was first discovered in reptiles as a 40-residue peptide isolated from leukocytes of the European pond turtle. Thus, based on its source, this type of AMP is known as turtle β-defensin 1 (TBD-1) [27]. Similarly, avian β-defensins include AvBD1-14 from the chicken, ostricacins from the ostrich (e.g., OSP-1 to OSP-4), and mallard duck β-defensins (AvBD2 and AvBD9), which are the common AMPs among avian family [27].

The most common mammalian AMPs belong to the members of the cathelicidin and defensins families. Mammalian cathelicidins are cationic peptides with an amphipathic structure in the form of α-helical, β-hairpin, or elongated conformations [26]. LL-37, the most well-studied cathelicidin, has an amphipathic structure, which can be modified into an aqueous solution to form an α-helix upon membrane interaction [26,28]. Mammalian defensins are classified into three sub-families: α, β, and θ. These subfamilies of defensins are synthesized first as prepropeptides, which share several features with mature peptides. These common features include cationic net charge (+1 to +11), short polypeptide sequences (18–45 amino acids), and three intramolecular disulfide bonds [27].

Based on their synthesis mechanisms, mammalian AMPs can be classified into either ribosomal-produced peptides [29] or non-ribosomal-produced peptides [30]. The synthesis of ribosomal AMPs occurs mainly in the cytoplasm of eukaryotic cells via the ribosome-dependent translation of genes encoding for AMPs, e.g., nisin [29,31]. By contrast, the synthesis of non-ribosomal AMPs is mediated by the peptide-synthesis-dependent mechanism in the cytosol of mammalian cells [25].

In contrast to ribosomal AMPs, the assembly of non-ribosomal AMPs contains not only the 20 common amino acids but also many rarer amino acids [32,33]. These different amino acids are synthesized by large enzymes, which are known as non-ribosomal peptide synthetases [33]. Non-ribosomal peptide synthetases are characterized by their ability to synthesize both cyclic and linear AMPs in the form of polypeptides, which give the AMPs their various molecular structures. [34]. A common example for both cyclic and linear AMPs is Gramicidin A (Figure 1), which appears as a small linear peptide with amphipathic and hydrophobic helices and a β-sheet secondary structure [35,36].

Cyclic peptides, such as polymyxin B (Figure 2A) [37], bacitracin (Figure 2B) [38], and vancomycin (Figure 2C) [39], are characterized by their unique amino acid compositions that appear in the form of lipopeptides or macrocyclic peptides. By contrast, peptides such as α defensin appear as bundles of α-helical rods in lipid bilayers (Figure 2D) [40].

While the cationic amphipathic helix is common in the secondary structure, particularly among bacteriostatic peptides [41], α-helical peptides are either hydrophobic or anionic with less selectivity towards microbes [42]. Apart from their different net charges, helical peptides are characterized by their ability to form hexameric clusters that can traverse bilayer membranes and surround an aqueous pore [43]. Consequently, the mechanism by which AMPs kill bacteria is mediated via the formation of pores, which leads to the disintegration of pathogen cell membranes [44,45]. The biological functioning of AMPs therefore depends on their ability to undergo structural modifications that allow them to interact with the membrane and elements of the cellular matrix.

## 3. Molecular Mechanisms of AMP Action

AMPs are characterized by their diverse activities and modes of action. These characteristics are determined by the type of target organisms and the mechanisms via which the AMPs exert their antimicrobial activity. For example, AMPs with antiviral activity are mostly associated with viral assembly, adsorption, and entry processes, in addition to their ability to target both RNA and DNA viruses. Among these antiviral peptides are indolizidine and human α-defensin 1 [46]. These AMPs have been shown to eliminate viruses via their incorporation into the viral envelope, leading to the instability of the virus assembly, and they subsequently deliver the viral entry into the host cell [44]. By contrast, AMPs such as lactoferricin have been shown to inhibit viral adsorption by binding to the specific viral receptors on the target cells [46]. Further, AMPs such as NP-1, an alpha-defensin that is derived from rabbit neutrophils, has been shown to inhibit viral assembly and maturation by binding the intracellular components that are essential for the cellular translocation of the virus in the host cell [46].

The most investigated AMPs are those with antibacterial activity. This type of AMP is characterized by its ability to interact with anionic bacterial membranes, leading to the disruption of the lipid bilayer [40].

Based on their molecular action, peptides with antimicrobial activity can be classified into two types. One of these types includes membrane-disrupting peptides, while the other one includes non-membrane-targeting peptides [40]. Although the molecular action of the main types of AMPs is different, some bacterial AMPs exert their activity via both membrane- and non-membrane-dependent mechanisms. Most AMPs trigger bacterial membrane destruction via interaction between their positively charged peptide molecules and the negatively charged cell surface as well as through hydrophobic interactions between the peptide amphipathic domain and membrane phospholipids.

Cationic AMPs have been demonstrated to exert their antibacterial activities via interaction with negatively charged bacterial membranes. The electrostatic interaction between cationic AMPs and the anionic components of the plasma membrane results in an increase in membrane permeability, and the release of AMPs into the cytoplasmic membrane, which subsequently, causes the lysis of the plasma membrane and, finally, the death of the microbial pathogen. To that end, four models have been proposed to describe the mechanisms whereby AMPs trigger the destruction of the microbial membrane. These include the barrel-stave (Figure 3A), toroidal pore (Figure 3B), carpet (Figure 3C), and aggregate (Figure 3D) models. In the barrel-stave model, the increased number of peptides binding to the membrane triggers membrane aggregation and conformational transformation. Consequently, the shift of local phospholipid head groups leads to cell membrane instability.

The barrel-stave mechanism is mediated via the vertical aggregation of helices into the lipid bilayer. The insertion of the transmembrane peptide bundle is organized in the cell membrane as staves of a barrel so that their hydrophobic face region is aligned with the central lipid region of the lipid bilayer. In parallel, the hydrophilic peptide constituents form the inner pore region that is filled with water [36]. The stable channels, namely, the barrel-like pores, which are formed in the cell membrane, allow the outflow of the cytoplasm. As consequence, the severe damage of the cell membrane results in cell collapse and, finally cell, death [33].

The toroidal pore model is mechanistically similar to the barrel-stave model; however, the mode of insertion of AMPs into the membrane and the binding behavior of AMPs with lipid molecules are different. In the toroidal pore model, the insertion of peptides into the membrane results in a continuous bending of the lipid monolayer from top to bottom [43]. The central water core is wrinkled with the inserted peptides and lipid head groups. Upon the formation of toroidal pores, the polar regions of the peptides start to line up with the lipid polar head groups.

To mediate their antimicrobial activities, AMPs first undergo confirmational modifications so that they can penetrate the phospholipid membrane. Following the penetration of the phospholipid membrane, the hydrophobic regions of the AMPs combine with the internal hydrophobic regions of the phospholipid bilayer, exposing the hydrophilic regions to the outside and subsequently increasing the membrane permeability of the microbial cell, which ultimately results in microbial death. Upon their entry into the cytoplasm, AMPs start to interfere with the intracellular components, leading to the dysregulation of cellular function via the mechanism mediated by the enhancement of DNA/RNA damage, inhibition of enzyme activity, and suppression of the transcription/translation processes, which are necessary for cell wall synthesis. Although in both barrel-stave and toroidal pore models, the mode of AMP insertion into the membrane determines the action of AMPs, in the carpet model, the action of AMPs depends on the concentration levels and electrostatic effect of the AMPs, as well as the net charge of the anionic component [43]. The hypothesis of the carpet model relies mainly on the initial aggregation of the peptides on the membrane in the monomeric or oligomeric form that ultimately cover the membrane as a carpet. As a consequence, the hydrophobic regions start to interact with the cell membrane while the hydrophilic ends face the aqueous solution. Once the concentration threshold has been reached, the aggregation of the peptides starts to enhance membrane permeability and ultimately membrane disruption [34]. Finally, in the aggregate model, the binding of the AMPs to the anionic cytoplasmic membrane causes the peptides and lipids to form a peptide–lipid complex micelle that opens membrane channels and allows the release of ions and intracellular contents, which ultimately leads to cell death [46]. In all models, the molecular action by which AMPs trigger microbial death depends on both a conformational change in the AMPs and the peptide–lipid ratio of the AMPs and the microbial membrane [46]. The conformational change in α-helical AMPs following anionic lipid membrane binding transforms the disordered structure of the AMPs in the aqueous solution into an amphiphilic α-helical structure, which facilitates the interaction of the AMPs with the microbial membrane [46]. Of note, in contrast to α-helical AMPs, AMPs with β-sheets are unable to undergo major conformational transitions during the interaction with the microbial membrane [46] due to the β-sheet AMPs’ stable disulfide bond bridges [16]. Peptide–lipid ratios likewise significantly impact conformational change and membrane lysis. At low peptide–lipid ratios, AMPs are located in parallel orientations on the surface of the plasma membrane [47], whereas at high peptide–lipid ratios, the AMPs becomes vertically oriented and are inserted into the hydrophobic center of the plasma membrane. This insertion of AMPs into the hydrophobic center of the plasma membrane increases the membrane permeability and subsequently enhances the release of both intracellular ions and metabolites that induce microbial cell death [48,49].

In addition to the destruction of the microbial membrane, AMPs have been reported to mediate their antimicrobial activity via intracellular-dependent mechanisms (Figure 3E). These include the induction of DNA/RNA damage, the inhibition of protein synthesis, enzyme activity, and the synthesis of a bacterial cell wall [50]. The above-mentioned AMP conformational changes and microbial membrane peptide–lipid ratios are unsurprisingly the main factors governing the ability of AMPs to pass through bacterial cell wall components, such as the lipopolysaccharides (LPSs) in the case of Gram-negative bacteria and lipoteichoic acid (LTA) and peptidoglycan in the case of Gram-positive bacteria [51]. AMP-mediated DNA/RNA damage has been found to be induced by the direct binding of AMPs to DNA or by the inhibition of DNA replication and transcription [48,49,52,53,54,55,56,57,58,59] AMPs such as Buforin II [52], a histone-derived antimicrobial peptide with a length of 21 amino acids, translocate across lipid membranes without affecting membrane permeability, and they trigger antimicrobial activity by binding to DNA/RNA [60]. Conversely, the AMP indolizidine, which displays antimicrobial activity against multi-drug resistance pathogens, has been shown to kill bacteria via the inhibition of DNA synthesis by penetrating membranes without inducing cell lysis. Anionic AMPs such as P2, isolated from Xenopus Leavis skin, was found to inhibit bacterial growth via interaction with microbial genomic DNA [61]. Other AMPs, such as PR-39, a proline/arginine-rich AMP isolated from the small intestine of pigs, has been reported to kill bacteria by penetrating the outer membranes of bacteria [62]. The entry of PR-39 into the cytoplasm was found to be associated with the inhibition of protein synthesis and acceleration of the ubiquitination of proteins, which are essential for DNA synthesis [63]. Proline-enriched AMPs have been reported to exert their antimicrobial activity through the interference of protein synthesis machinery by binding to ribosomes [63]. The N-terminal (1–25) and (1–31) residues of the non-lytic proline-rich AMP (PrAMP) Bac 5, for example, bind to the tunnel of ribosomes and prevent the translation process [64]. The proline-enriched AMP oncocin inhibits mRNA translation by binding the 70S ribosome, whereas apidaecin inhibits 50s ribosome assembly [65]. Api137, an apidaecin-derived peptide, binds to the ribosomes of *E. coli* and trap release factors 1 (RF1) or 2 (RF2) to trigger translation termination [66].

The inhibition of microbial pathogens’ intracellular enzymes has also been reported to be a mechanism through which some AMPs exert their antimicrobial activity. Pyrrhocoricin binds to the bacterial heat shock protein DnaK and subsequently inhibits ATPase action [25]. Microcin J25, a ribosomal synthesized and post-translationally modified AMP, binds to the secondary channel of the RNA polymerase, blocking the entry of substrates through the channel [61]. LL-37 inhibits the activity of palmitoyl transferase PagP [57,58]. Pag P is an enzyme located in the outer membrane of Gram-negative bacteria and facilitates membrane permeability via activated lipid A acylation [67]. Finally, NP-6, isolated from Sichuan pepper seeds, inhibits *E. coli* β-galactosidase activity [67].

The inhibition of bacterial cell wall synthesis is a common therapeutic strategy to treat pathogenic bacteria infection. The anti-leishmanial drug candidate, human neutrophil peptide-1 (HNP-1), inhibits bacterial cell wall synthesis by penetrating the outer and inner membranes of *E. coli* and suppresses the synthesis of DNA, RNA, and proteins [68]. HNP1′s antimicrobial activity is mediated by its interaction with lipid II. HNP1 binds to a highly conserved non-peptide motif of peptidoglycan precursor (lipid II) and teichoic acid precursor (lipid III) [61], resulting in the inhibition of cell wall synthesis and subsequent lysis. HNP1 has excellent activity against a wide range of Gram-positive bacteria, including multi-drug resistant organisms, such as Methicillin-resistant *Staphylococcus aureus* (MRSA), Vancomycin Intermediate *S. aureus* (VISA), Vancomycin-resistant enterococci (VRE), Clostridium difficile, *Streptococcus pneumoniae*, and *Mycobacterium tuberculosis* [23]. Finally, teixobactin is a cyclic dipeptide containing an unusual amino acid, enduracididine [68]. This AMP is a member of a new class characterized by their specific action on unique targets in cell wall synthesis.

## 4. Therapeutic Potential of Antimicrobial Peptides

Human infections are typically polymicrobial and stem mainly from oral infections, surgical wounds, diabetic foot ulcers, cystic-fibrosis-related lung infections, urinary tract infections, and otitis media infections [69,70,71]. Therefore, the treatment of polymicrobial infection is more challenging when compared with monomicrobial infections. In contrast to traditional antibiotics, AMPs are characterized by their ability to target both monomicrobial and polymicrobial infections without the development of cross-resistance [72]. Thus, the advantage of AMPs over traditional antibiotics is their ability to act directly on the bacterial membrane when compared to their indirect action on the intracellular targets. Other advantages of AMPs over conventional antibiotics involve the actions mediated by their different characteristics, including their ability to function against both antibiotic-resistant and -sensitive microbial pathogens and their ability to target monomicrobial and polymicrobial infections without the development of cross-resistance [64,73]. However, the therapeutic success of AMPs in the treatment and prevention of bacterial infection may result from their ability to act directly on the bacterial membrane, rather than their indirect action on intracellular targets [74]. Furthermore, the ability of a single AMP to exert its antimicrobial activity via multiple mechanisms, and through different pathways [75], suggests the clinical relevance of AMPs in the treatment and prevention of microbial pathogens. However, the establishment of novel, clinically relevant therapeutic approaches that target multiple pathogens in mixed populations, thereby replacing traditional antibiotics, is tangible.

The production of endogenous AMPs by multicellular organisms constitutes a host defense mechanism against pathogenic microbes. Based on the broad spectrum of their antimicrobial activity, AMPs are promising therapeutic agents for infection control [25]. In addition to their antimicrobial activities against various pathogens, including bacteria (Gram-positive and Gram-negative bacteria), fungi, and viruses [76], many AMPs are effective against multi-drug resistant (MDR) bacteria and have low propensity for the development of resistance [77]. AMPs are also involved in the promotion and regulation of the innate immunity system [78]. Finally, the use of AMPs against biofilm formation has been widely reported over the last few decades [79]. Many AMPs kill cells in biofilms and inhibit biofilm formation via the interference with the abundant extracellular polymeric substances (EPSs) of microbial cells. These EPSs are known to be functionally responsible for the protection of microbial cells from the surrounding environment [80].

Despite the potential therapeutic benefits of AMPs when compared with existing antibiotics, AMPs have some limitations that hinder their development for clinical use [81]. Most natural AMPs are characterized by poor absorption, distribution, metabolism, and excretion, in addition to their short half-life and low permeability and solubility [82]. Moreover, AMPs have a high production cost and a degree of toxicity, particularly in oral administration. All these properties are considered as major hindrances for the development of novel AMP-based treatments. To overcome these AMP limitations that hamper clinical application, several studies are urgently needed to improve the functional properties of AMPs, such as their absorption, distribution, metabolism, excretion, cytotoxicity, and proteolytic stability. An improvement in the functional properties of AMPs may involve the alteration of the peptide composition and the modification of their post translation of AMPs.

To that end, several technical procedures have been proposed to improve the functional properties of AMPs. These include the modification of the chemical structure of AMPs via the introduction of unusual amino acids, such as D-form amino acids, or by the acetylation or amidation of the terminal regions of AMPs. As widely reported, the modification of the chemical structure of AMPs was found to improve the stability of their peptides and prevent their proteolytic degradation [83]. Similarly, the delivery of AMPs using liposome encapsulation was found to preserve the stability of AMPs and to reduce their toxicity [84].

AMPs are an essential component of the innate skin defense mechanisms and are considered to be a first-line barrier providing protection against microbial pathogens [85]. AMPs are closely associated the with innate skin immunity and are known to regulate immunity by interacting with various immune cells and linking innate and adaptive immune responses during infection. These AMPs include, β-defensins (BD) [86], cathelicidins (human hCAP18/LL37) [87], RNase 7 [88], and secretory leukocyte protease inhibitor (SLPI) [89]. Apart from their significant role in the regulation of innate skin immunity, AMPs such as defensins and cathelicidins have also been reported to play a key role in the regulation of the innate immunity of the lung [86,90]. To that end, both defensins and cathelicidins belong to a family of AMPs, which are mostly detected in the secretion of airways [87]. The exogenous administration of defensins and cathelicidins has been reported as an effective strategy in the prevention and treatment of infection. In particular, tachyplesin III, a β-sheet peptide isolated from the hemocytes of the horseshoe crab, has been evaluated for antimicrobial activity in lung polymicrobial co-infection pneumonia [89,91].

Naturally produced AMPs in the oral cavity play key roles in the maintenance of microbial homeostasis and oral cavity health stasis [91,92]. These AMPs are characterized by their antimicrobial activity against oral bacteria, which has been evaluated against oral infections, as widely reported in several studies [91,92]. D-Cateslytin (D-Ctl), an AMP derived from L-Cateslytin, has been observed to have therapeutic potential against bacterial infection in combination with several antimicrobials [92,93] and has been reported to be an antifungal agent in the treatment of oral infections associated with Candida albicans [94].

The most important advantages of AMPs over conventional therapeutics are attributed to the potential of AMPs to offer innovative and effective solutions to the treatment of mixed populations with polyinfections and to differentiate between pathogenic bacteria and protective normal flora. Therefore, the development and evaluation of AMPs with the ability to target multiple pathogens in mixed populations without the destruction of the protective normal flora represents an important public health issue.

## 5. Conclusions

AMPs are characterized by their broad spectrum of antimicrobial activities and are powerful regulators of innate immunity. AMPs have a strong cell-killing efficiency on microbial pathogens, particularly MDR bacteria. In addition, AMPs offer an alternative approach to overcome the antibiotic resistance of most microbial pathogens. Although AMPs may be able to overcome the limitations of current antibiotics due to their antimicrobial activity, their shortcomings include poor stability, toxicity, and unexplored adverse effects, which limit their clinical application. However, continued development and evaluation of functional AMPs may allow for the modification of natural AMPs, thereby facilitating the production of new AMPs with clinically desirable characteristics. Some AMPs have been approved for clinical application, while others remain under investigation in clinical trials. Therefore, the development and evaluation of AMPs with the ability to target multiple pathogens in mixed populations without the destruction of the protective normal flora represents an exciting antimicrobial therapeutic strategy.

## Figures and Tables

**Figure 1 pharmaceutics-15-00072-f001:**
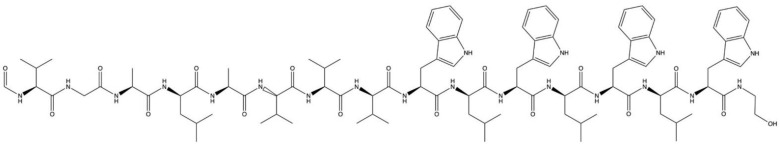
Chemical structure of antimicrobial peptide gramicidin A. Gramicidin A is a linear antimicrobial peptide and one of the three known gramicidins (A, B, and C). They are non-ribosomal peptides that consist of the following 15 L- and D-amino acids: formyl-L-X-Gly-L-Ala-D-Leu-L-Ala-D-Val-L-Val-D-Val-L-Trp-D-Leu-L-Y-D-Leu-L-Trp-D-Leu-L-Trp-ethanolamine. The difference between gramicidins A, B, and C is that the amino acid position Y is L-tryptophan in gramicidin A, L-phenylalanine in B, and L-tyrosine in C. The isoforms of the gramicidins A, B, and C are determined by the existence of L-valine or L-isoleucine at position X of anion acid and the origin [10].

**Figure 2 pharmaceutics-15-00072-f002:**
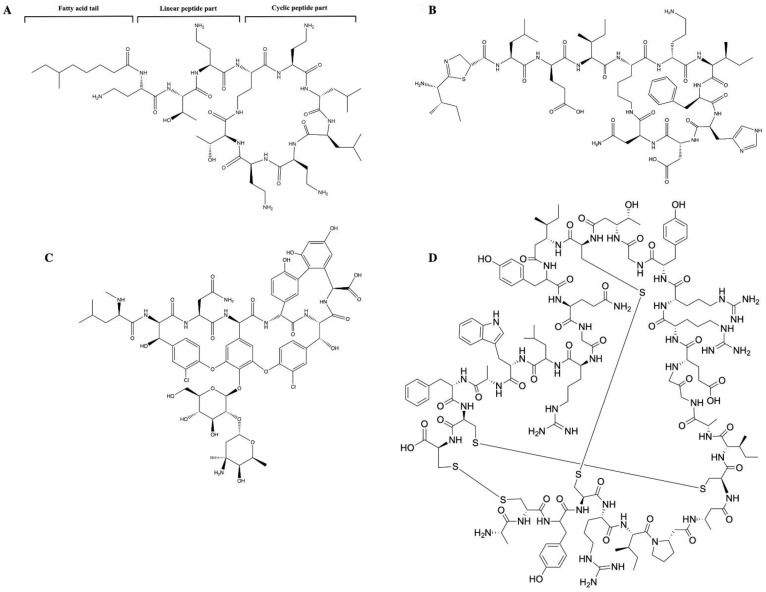
Chemical structure of cyclic antimicrobial peptides: (**A**) Structure of polymyxin B, including fatty acyl tail, linear peptide, and cyclic peptide (**B**) Chemical structure of bacitracin: bacitracin is an AMP that consists of D-aspartic acid, D-phenylalanine, D-ornithine, D-glutamic acid, and a ring of thiazoline containing amino acids. (**C**) Chemical structure of vancomycin: vancomycin is a branched tricyclic glycosylated non-ribosomal peptide. (**D**) Chemical structure of defensin: defensins comprise an N-terminal β-strand followed by an α-helix and two more β-strands. The β-strands form a triple-stranded antiparallel β-sheet that can be stabilized by disulphide bonds. Two of the disulphide bonds connect the α-helix and the central β-strand, while a third disulphide bond stabilizes the structure by linking the β-strand.

**Figure 3 pharmaceutics-15-00072-f003:**
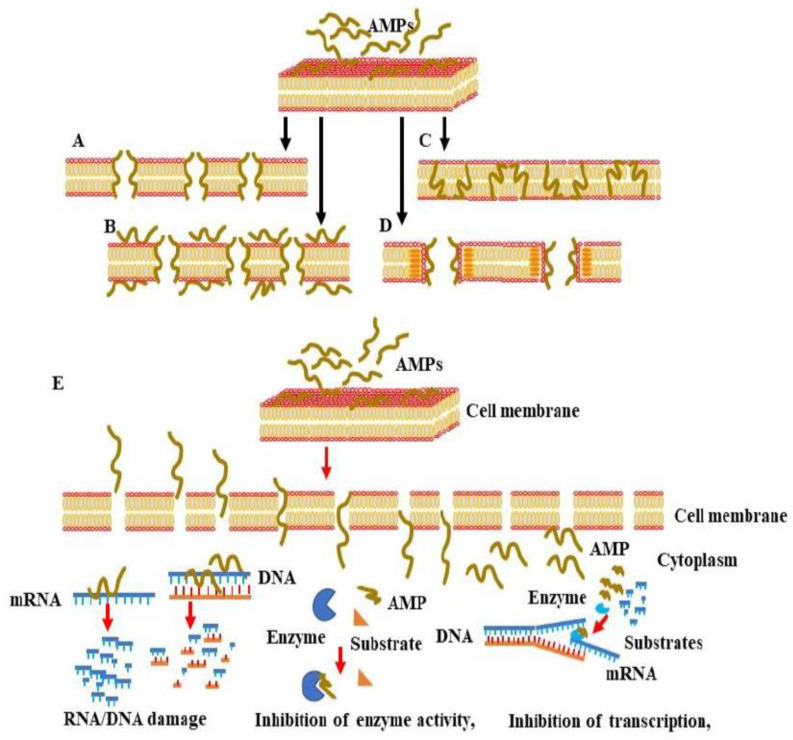
Proposed models for AMP-induced membrane permeability, membrane penetration, and interference with cellular components. AMPs exert their antimicrobial activity via interaction with negatively charged membranes to mediate and rapidly increase membrane permeability, cell membrane lysis, or the release of intracellular contents, leading to microbial cell death. There are four main models of membrane pore formation, namely, the barrel-stave model (**A**), toroidal pore model (**B**), carpet model (**C**) and aggregate model (**D**). (**E**) Mechanisms of the penetration of AMPs into the cytoplasm of the microbial cell and interference with intracellular components.

## Data Availability

The data that support the findings of this study are available from the corresponding author upon reasonable request.

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
