# Peer review of "Antimicrobial Proteins: Structure, Molecular Action, and Therapeutic Potential"

_pharmaceutics, 2022, doi:10.3390/pharmaceutics15010072_

Round 1

Reviewer 1 Report

Thanks to the authors to this nice review, but I have some comments

Gm +ve and Gm -ve should be written in correct way

all microorganisms should be written in italic

the chemical structures should be drawn using ChemDraw

the references should be revised according to the journal style

Author Response

enclosed find Authors response  to Reviewer 1

Reviewer 2 Report

The manuscript in its current format suffers from poorly and not correctly constructed phrases and as such the content can not be properly evaluated.

All chemical structures should be drawn using a drawing software (like chemdraw) and all structures should be drawn using the same object setting.

Author Response

Authors response to the coment of Reviewer 2

Reviewer 3 Report

The article does not make any original contribution, although it addresses a promising topic from a scientific point of view, that of antimicrobial peptides as therapeutic alternatives.
Although in the abstract the authors specify "This review provides a current overview into the structure, molecular action, and therapeutic potential of AMPs", these detailed aspects are not found in the article.
The article contains many unfinished sentences, incompletely or wrongly argued ideas, incorrect scientific terms (eg viral membranes).
in the text of the article I emphasized all these aspects and you can find them in the attached document.
Also, based on the long list of errors I discovered, the article seems unprofessionally written and without serious and in-depth documentation. In this context, I strongly recommended to reject the article.

Author Response

Authors response to reviewer 3

Round 2

Reviewer 2 Report

The authors did extensive revision of their manuscript and it reads better now. I have just minor corrects that I have made within the manuscript. See attachment for this

Author Response

The required changes have been done and undelined over all in the  main text of the manuscript.

Many thanks

Reviewer 3 Report

Dear Editor

I agree with the changes but the text requires moderate English language and style spell check and the writing of scientific names in italics (for example Clostridium difficile - line 382).

Author Response

(The authors gave the same response as above.)
